# Association between COVID-19 and Seasonal Influenza Vaccines to Vaccine Hesitancy, Intolerance of Uncertainty and Mental Health

**DOI:** 10.3390/vaccines11020403

**Published:** 2023-02-09

**Authors:** Maayan Shacham, Yaira Hamama-Raz, Menachem Ben-Ezra, Yafit Levin

**Affiliations:** 1Unit of Medical Education, Department of Oral Rehabilitation, Goldschleger School of Dental Medicine, Tel Aviv University, Tel Aviv 6997801, Israel; 2School of Social Work, Faculty of Humanities and Social Sciences, Ariel University, Ariel 40700, Israel; 3School of Education, Faculty of Humanities and Social Sciences, Ariel University, Ariel 40700, Israel

**Keywords:** vaccine hesitancy, COVID-19, influenza, pandemics, intolerance of uncertainty, mental health

## Abstract

Vaccine hesitancy is a universal problem that is becoming more prevalent, ranging from partial acceptance to the complete refusal of various vaccines. The current study seeks to assess the relationship between vaccine hesitancy, intolerance of uncertainty, and mental health factors and those who were vaccinated against COVID-19 and seasonal influenza in comparison to those who did not vaccinate against both or decided to be vaccinated with only one of these vaccines. Employing a cross-sectional design, 1068 Israeli participants were recruited via social media (mainly Facebook) and Whatsapp and completed questionnaires assessing vaccine hesitancy, intolerance of uncertainty, and mental health factors. Our results revealed that previous history of neither COVID-19 nor seasonal influenza vaccination was associated with increased vaccine hesitancy. In addition, individuals who received either one vaccine or both claimed elevated levels of intolerance of uncertainty and reported elevated levels of mental health symptoms. Therefore, an association between vaccine hesitancy and intolerance of uncertainty and mental health symptoms is demonstrated. Future campaigns against vaccine hesitancy may focus on the intolerance of uncertainty in vaccine-hesitant individuals.

## 1. Introduction

Vaccine hesitancy seems to have gained increased interest since the emergence of the severe acute respiratory syndrome coronavirus 2 (SARS-CoV-2) virus and the COVID-19 pandemic [1]. It is known that individuals may present with various levels of vaccine hesitancy [2], which may be explained via different theories, namely, the health belief model (HBM) [3,4] and the terror management health model (TMHM; [5,6]). HBM encompasses several domains, including perceived susceptibility, severity, benefits, and barriers to a health issue, as well as self-efficacy to engage in a particular behavior towards such issues [3]. The HBM domains were previously found to play an important role in predicting cooperation with influenza and COVID-19 vaccines [3,7,8,9,10].

TMHM is based on terror management theory, which postulates that every human may choose to utilize either proximal or distal defenses to deal with existential concerns regarding health issues, which are often parallel to death issues [5,6]. According to the TMHM, as death issues become more conscious, decisions regarding health are mainly decided by proximal defenses in order to reduce the perceived vulnerability to that health threat and, consequently, concerns about possible death. On the other hand, if death awareness is active in cognition but is outside of proximity, then distal defenses will guide health-related decisions in order to keep safe the symbolic value of the self [6]. In the case of influenza and SARS-CoV-2 pandemics, THMH is further supported as it seems that the risk of dying from a viral infection may lead to a reduction in distal defenses, which are thought to play an integral role in maintaining the symbolic value of the self [6,11,12,13]. In relation to the COVID-19 vaccine, TMTH was shown to influence perceived susceptibility (by increasing healthy behaviors and reducing death anxiety [14]) as well as to increase vaccine hesitancy [15]. Based on the described theoretical background, we aimed to provide further assessment of the relationships between COVID-19 and seasonal influenza vaccines and vaccine hesitancy and related psychological factors, namely, intolerance of uncertainty and mental health.

Vaccine hesitancy is often explored in the scientific literature by addressing anti-vaccination attitudes, utilizing specific scales [16,17,18]. Such scales address several domains with regard to vaccine hesitancy, such as trust or mistrust of vaccine benefits, worries over unforeseen future effects, concerns about commercial profiteering, and preference for natural immunity [16,17,18]. In order to aid public health policymakers, it is crucial to understand the attitudes toward vaccines that serve as the basis for vaccine hesitancy. Based on such understanding, health campaigners can tailor more specific campaigns in order to better address such issues. During the COVID-19 pandemic, increased vaccine hesitancy towards the COVID-19 vaccine was observed in the adult Israeli population [18]. In addition, recent studies also indicated that past influenza vaccinations positively predicted the intentions to be vaccinated against COVID-19 [19,20]. Prior to the COVID-19 pandemic, a systemic review regarding influenza vaccine hesitancy highlighted the importance of addressing psychological factors related to vaccination attitudes [21]. Intolerance of uncertainty and mental health might serve as these psychological factors.

Evidence points to a relationship between intolerance of uncertainty and mental health due to fear of SARS-CoV-2 viral infection [22,23]. Intolerance of uncertainty (namely, an individual’s negative emotions, cognitions, and behaviors when uncertainty is experienced [24]) was found to be associated with depression during COVID-19 [25] and to predict vaccine hesitancy in a non-anxious population [19]. The impact on mental health during the COVID-19 pandemic has been widely addressed [26], along with elevated psychological impacts since the introduction of COVID-19 vaccines, mainly anxiety and vaccine hesitancy [27].

Nevertheless, the relationship between vaccine hesitancy, intolerance of uncertainty, and mental health factors and those who were vaccinated against COVID-19 and seasonal influenza in comparison to those who did not vaccinate against both or decided to be vaccinated with only one of these vaccines remains uninvestigated. Although some studies have investigated the relationship between psychological factors and vaccine hesitancy since the COVID-19 pandemic [23,28,29], these did not address the interplay between the aforementioned factors, as previously depicted [30]. Shedding light on these relationships will help policymakers design appropriate interventions to reduce vaccine hesitancy. As previously stated, the majority of studies conducted on vaccine hesitancy were performed on healthcare worker populations [21], and several recent studies had small sample sizes [14,15,31], thus further necessitating studies on large general population samples. Therefore, the current study aims to address the aforementioned in an adult Israeli population.

## 2. Materials and Methods

### 2.1. Recruitment and Eligibility

Data were collected from 6 November to 7 December 2022. Eligibility criteria specified that participants should be: aged 18 or over; Israeli residents at the time the survey was conducted; able to give informed consent; fluent in the native language.

### 2.2. Sample Size

As a minimum, we estimated that 599 participants would be required to detect low–medium effect sizes of 0.20, with 90% power and a 5% significance level. The study sample size was n = 1068.

### 2.3. Sampling and Procedures

The study was conducted according to the STROBE guidelines for observational studies. The STOBE guidelines are STrengthening the Reporting of OBservational studies in Epidemiology. These guidelines are considered the common practice for good epidemiological studies; see https://www.strobe-statement.org/ (accessed on 1 September 2022) for further information. We used the Qualtrics platform to deploy the survey. The Qualtrics platform is a widely used survey platform for conducting online surveys. Its ease of usage is convenient for both research teams and sample populations. It is a widely used platform, utilized by numerous academic institutions worldwide. The participants were approached using emails, social media (mainly Facebook), and Whatsapp. All the participants signed an electronic informed consent form. The study was approved by Ariel University’s ethics committee.

### 2.4. Measurements

Demographic variables: The demographic variables were age (mean = 29.82; SD = 24.00; range 18–81); sex, with men coded as “1” and women as “2” (68.2% of the sample, *n* = 728). Most of the participants were not in a committed relationship (58.7% of the sample, *n* = 627), coded as “1” for not being in a committed relationship and “2” for being in a committed relationship. Education was defined as the mean years of study (mean = 13.73; SD = 2.49; range 7–24).

Vaccine-related questions: The participants were asked if they were vaccinated against COVID-19 (1 = no [*n* = 73; 7.8%]; 2 = yes [*n* = 860; 92.2%]) and against influenza viruses (1 = no [*n* = 814; 86.2%]; 2 = yes [*n* = 130; 13.8%]). This combination created four subgroups (0 = no vaccines = 70; 7.5%; 1 = vaccine for influenza only = 3; 0.003%; 2 = vaccine for COVID-19 solely = 735; 78.9%; 3 = both vaccines = 123; 13.2%). As the group for the influenza vaccine was extremely low, we aggregated it into the COVID-19 vaccine group. Hence, we have three groups: 0 = no vaccine = 70; 7.5%; 1 = COVID-19 or influenza vaccine = 738; 79.3%; 2 = both vaccines = 123; 13.2%).

Vaccine hesitancy was measured by the Vaccination Attitudes Examination Scale (VAX). The VAX scale is a 12-item scale used to assess anti-vaccination attitudes [18]. Each item is scored on a scale of “1 = strongly disagree” to “6 = strongly agree”. A higher score indicates a higher level of anti-vaccination attitude. The internal consistency in the current study was found to be good (Cronbach’s α = 0.90).

Intolerance of uncertainty was measured by the Intolerance of Uncertainty Scale-12 [24]. This measure consists of 12 items (e.g., “It frustrates me not having all the information I need”) and utilizes a 5-point Likert scale, from 1 (“Not at all characteristic of me”) to 5 (“Very characteristic of me”) to evaluate two sub-scales of IU (prospective anxiety and inhibitory anxiety). A greater score in each subscale indicates a greater level of prospective anxiety and inhibitory anxiety. The internal consistency in the current study was found to be good (Cronbach’s α = 0.89).

Mental health was measured by the Patient Health Questionnaire-4 (PHQ-4) [32]. The PHQ-4 begins with the stem question: “Over the last 2 weeks, how often have you been bothered by the following problems?”. Responses are rated on a 4-point Likert scale, ranging from 0 (“not at all”) to 3 (“nearly every day”). The PHQ-4 total score ranges from 0 to 12, with higher scores indicating poorer mental health. The internal consistency in the current study was found to be good (Cronbach’s α = 0.81).

### 2.5. Statistical Analysis

The analytic plan included three steps. First, we conducted a correlation matrix of all the study variables. Second, we conducted a one-way ANOVA comparing the three vaccine groups (0—no vaccines, 1—one vaccine, 2—two vaccines) on the following variables: age, sex, relationship status, years of education, vaccine hesitancy (VAX-12), intolerance of uncertainty (IUS-12), and mental health (PHQ-4). Post-hoc comparisons were conducted using the Bonferroni post-hoc test. Third, we conducted a MANCOVA with the vaccine group as an independent factor and age, sex, relationship status, and years of education as covariates. The dependent variables were vaccine hesitancy (VAX-12), intolerance of uncertainty (IUS-12), and mental health (PHQ-4).

## 3. Results

In step 1, we present the basic correlation matrix results relevant to the current study. The vaccine group was negatively correlated with vaccine hesitancy (VAX-12) (r = −0.129; *p* < 0.001), intolerance of uncertainty (IUS-12) (r = −0.152; *p* < 0.001), and mental health (PHQ-4) (r = −0.163; *p* < 0.001). For more information, please see Table 1.

In step 2, we present the results of the one-way ANOVA with the vaccine group as an independent variable, along with post-hoc comparisons for the study variables. Age differed among the vaccine’s groups (no vaccines = 31.53 [14.03]; one vaccine = 28.33 [10.91]; two vaccines = 37.96 [20.01]; F = 30.87; *p* < 0.001; post-hoc Bonferroni G3 > G2; G3 > G1). Similar results were found for mean years of education (no vaccines = 13.98 [2.51]; one vaccine = 13.54 [2.28]; two vaccines = 15.23 [3.12]; F = 23.05; *p* < 0.001; post-hoc Bonferroni G3 > G2; G3 > G1) and those who were not in a committed relationship (no vaccines = 58.6%; one vaccine = 60.4%; two vaccines = 47.2%; Jonckheere–Terpstra test statistic = 2.114; *p* < 0.05). Beyond demographics, we found similar patterns with regard to the dependent variables, namely, vaccine hesitancy (VAX-12) (no vaccines = 58.23 [10.65]; one vaccine = 37.07 [11.13]; two vaccines = 29.64 [10.45]; F = 152.34; *p* < 0.001; post-hoc Bonferroni G1 > G2; G1 > G3; G2 > G3), intolerance of uncertainty (IUS-12) (no vaccines = 25.35 [7.59]; one vaccine = 28.52 [9.42]; two vaccines = 28.17 [9.55]; F = 3.29; *p* < 0.05; post-hoc Bonferroni G2 > G1), and mental health (PHQ-4) (no vaccines = 1.37 [1.78]; one vaccine = 2.24 [2.45]; two vaccines = 1.99 [2.29]; F = 3.98; *p* < 0.05; post-hoc Bonferroni G2 > G1). For more information, see Table 2.

In step 3, we present the result of the multivariate tests of the study variables, with demographics and vaccine group as the independent variables and vaccine hesitancy (VAX-12), intolerance of uncertainty (IUS-12), and mental health (PHQ-4) as the dependent variables. All the independent variables were found to be significant, namely, age (Wilks’ Lambda value = 0.966; F = 8.724; *p* < 0.001; partial η2 = 0.034), sex (Wilks’ Lambda value = 0.950; F = 13.100; *p* < 0.001; partial η2 = 0.050), relationship status (Wilks’ Lambda value = 0.986; F = 3.410; *p* < 0.05; partial η2 = 0.014), years of education (Wilks’ Lambda value = 0.969; F = 7.839; *p* < 0.001; partial η2 = 0.031), and, most significantly, vaccine group (Wilks’ Lambda value = 0.728; F = 42.628; *p* < 0.001; partial η2 = 0.147). For more information, see Table 3.

In step 4, we present the MANCOVA results. Age was found to be significantly associated with each of the dependent variables, namely, VAX-12 (F = 5.918; *p* < 0.05; partial η2 = 0.008), IUS-12 (F = 13.831; *p* < 0.001; partial η2 = 0.018), and PHQ-4 (F = 10.611; *p* < 0.001; partial η2 = 0.014). Sex presented a similar pattern for VAX-12 (F = 17.885; *p* < 0.001; partial η2 = 0.023), IUS-12 (F = 18.181; *p* < 0.001; partial η2 = 0.024), and PHQ-4 (F = 15.866; *p* < 0.001; partial η2 = 0.021), and vaccine group for VAX-12 (F = 125.116; *p* < 0.001; partial η2 = 0.251), IUS-12 (F = 5.473; *p* < 0.05; partial η2 = 0.014), and PHQ-4 (F = 3.982; *p* < 0.05; partial η2 = 0.011). Relationship status was only significant with regard to PHQ-4 (F = 8.644; *p* < 0.01; partial η2 = 0.011) and years of education with regard to VAX-12 (F = 22.631; *p* < 0.001; partial η2 = 0.029). See Table 4 for more information.

## 4. Discussion

The current study has assessed the associations between COVID-19 and seasonal influenza vaccines and vaccine hesitancy, intolerance of uncertainty, and mental health. Our results indicate that the number of vaccinations is significantly associated with these factors. Specifically, those who had no history of either COVID-19 or seasonal influenza vaccinations reported elevated levels of vaccine hesitancy. Such findings have been reported in previous studies [19,20] and may be explained via the HBM and THMH theories. As risk perception rises, those who demonstrate anti-vaccine attitudes will presumably opt to be unvaccinated as their way of feeling safer. In the case of THMH theory, it may be that those individuals keep their distal defenses active by staying away from what they perceive as the risk, which may be the vaccine itself rather than the viral infection. As previously reported, the adult Israeli population showed elevated levels of COVID-19 vaccine hesitancy during its earliest introduction to the population [18]. Thus, it seems that vaccine hesitancy in the adult Israeli population seems to be elevated and, thus, may require Israeli health policymakers to adjust vaccine campaigns in order to reduce levels of vaccine hesitancy.

Another interesting result was related to the intolerance of uncertainty. There are two approaches to interpreting and analyzing IUS-12 scale scores. The first is calculating each subscale (prospective and inhibitory) and using them separately. We did not deploy this method in the current paper. For the purpose of the current study, we utilized and scored the IUS-12 as a whole, meaning a global measure of intolerance to uncertainty. We found that those who received either one vaccine or both vaccines reported elevated levels of intolerance of uncertainty. This is consistent with the HBM as those who report elevated levels of intolerance of uncertainty may engage in the pro-active monitoring of their health [33,34], namely, receiving a vaccine in order to prevent viral infection. That is, their perceived risk of getting infected is higher than the perceived risk of being vaccinated against it. However, such results were not replicated in a recent study [35], presumably due to different methodologies.

As to mental health outcomes, we revealed that the individuals who had received either of the vaccines exhibited elevated levels of mental health outcomes. Such results were previously demonstrated in the COVID-19 pandemic [22,23,25,27]. Based on the THMH theory, it may be that the individuals who demonstrate a perceived risk of viral infection and utilize distal defenses to get vaccinated are also exposed to an increased impact on their own mental health.

In addition, younger age was found to be associated with being vaccinated with both vaccines; that is, less vaccine hesitancy was found among younger individuals. Such data were also reported in a recent review study [1]. Females tended to be in the majority in all vaccine groups. The latter is not surprising as the majority of the current study participants were female, serving as one limitation of the current study. Additional limitations of the current study may be its cross-sectional study design, its short period of time (one month), being online, and being deprived of certain demographics. Since the number of individuals who were vaccinated solely for seasonal influenza was extremely low (n = 3), which led to the aggregation of the vaccine groups, future studies may address this issue and compare it to the results of the current study. However, the current study’s strengths are its sample size and the collection and demonstration of the interplay between vaccine hesitancy, intolerance of uncertainty, and mental health outcomes based on a history of seasonal influenza and COVID-19 vaccinations.

As previously noted, vaccine hesitancy seems to be important during the COVID-19 pandemic as its levels are rising [1]; this is further supported by the results of the current study. Our study demonstrates the relationship between vaccine hesitancy and intolerance of uncertainty and mental health outcomes. Therefore, the results of the current study may be utilized by health policymakers to address campaigns against vaccine hesitancy by focusing on its potential mental health outcomes, such as depression and anxiety, and suggesting ways to improve the intolerance of uncertainty in those individuals who possess anti-vaccine attitudes.

## Figures and Tables

**Table 1 vaccines-11-00403-t001:** Bivariate correlations among the study variables (*n* = 1068) ^a^.

	1	2	3	4	5	6	7	8
1. Age								
2. Sex (female)	−0.063 *							
3. Relationship status (not in a committed relationship)	0.425 ***	0.047						
4. Years of education	0.546 ***	0.06	0.436 ***					
5. Vaccine group	0.160 ***	−0.032	0.068 *	0.156 ***				
6. Vaccine hesitancy (VAX-12)	−0.014	0.143 ***	−0.025	−0.129 ***	−0.450 ***			
7. Intolerance of uncertainty (IUS-12)	−0.214 ***	0.174 ***	−0.111 ***	−0.152 ***	0.05	0.054		
8. Mental health (PHQ-4)	−0.216 ***	0.138 ***	−0.181 ***	−0.163 ***	0.036	0.047	0.407 ***	

* *p* < 0.05; *** *p* < 0.001. ^a^ = Valid n ranged from 772 to 1068.

**Table 2 vaccines-11-00403-t002:** One-way ANOVA of the study variable by vaccine group.

Variables	Vaccine Group Statistics				
	No Vaccines (*n* = 70)	One vaccine (*n* = 738)	Two vaccines (*n* = 123)	One-Way ANOVA	Post-hoc tests
Age, Mean (SD)	31.53 (14.03)	28.33 (10.91)	37.96 (20.01)	F = 30.87 ***	G3 > G2; G3 > G1
Sex (female), *n* (%)	51 (72.9%)	504 (68.3%)	81 (65.9%)		Jonckheere–Terpstra Test statistic = −0.959
Relationship status (Not in a committed relationship), *n* (%)	41 (58.6)	446 (60.4)	58 (47.2)		Jonckheere–Terpstra Test statistic = 2.114 *
Years of education, Mean (SD)	13.98 (2.51)	13.54 (2.28)	15.23 (3.12)	F = 23.05 ***	G3 > G2; G3 > G1
Vaccine hesitancy (VAX-12), Mean (SD)	58.23 (10.65)	37.07 (11.13)	29.64 (10.45)	F = 152.34 ***	G1 > G2; G1 > G3; G2 > G3
Intolerance of uncertainty (IUS-12), Mean (SD)	25.35 (7.59)	28.52 (9.42)	28.17 (9.55)	F = 3.29 *	G2 > G1
Mental health (PHQ-4), Mean (SD)	1.37 (1.78)	2.24 (2.45)	1.99 (2.29)	F = 3.98 *	G2 > G1

* *p* < 0.05. *** *p* < 0.001.

**Table 3 vaccines-11-00403-t003:** Multivariate tests of the study variable by vaccine group.

Variable	Wilks’ Lambda Value	F	Partial Eta Squared	Observed Power
Age	0.966	8.724 ***	0.034	0.995
Sex	0.950	13.100 ***	0.050	1.000
Relationship status	0.986	3.410 *	0.014	0.769
Years of Education	0.969	7.839 ***	0.031	0.990
Vaccine group	0.728	42.628 ***	0.147	1.000

* *p* < 0.05. *** *p* < 0.001.

**Table 4 vaccines-11-00403-t004:** MANCOVA results of the study variables.

Factor	Variable	Sum of Squares	Mean Square	F	Partial η^2^	Observed Power
Age	VAX-12	701.213	701.213	5.918 *	0.008	0.681
	IUS-12	1117.946	1117.946	13.831 ***	0.018	0.960
	PHQ-4	55.388	55.388	10.611 ***	0.014	0.902
Sex	VAX-12	2119.213	2119.213	17.885 ***	0.023	0.988
	IUS-12	1469.569	1469.569	18.181 ***	0.024	0.989
	PHQ-4	82.814	82.814	15.866 ***	0.021	0.978
Relationship status	VAX-12	123.850	123.850	1.045	0.001	0.175
	IUS-12	63.031	63.031	0.780	0.001	0.143
	PHQ-4	45.119	45.119	8.644 **	0.011	0.836
Years of education	VAX-12	2681.559	2681.559	22.631 ***	0.029	0.997
	IUS-12	141.149	141.149	1.746	0.002	0.262
	PHQ-4	3.554	3.554	0.681	0.001	0.131
Vaccine group	VAX-12	29,649.899	14,824.949	125.116 ***	0.251	1.000
	IUS-12	884.818	442.409	5.473 **	0.014	0.849
	PHQ-4	41.572	20.786	3.982 *	0.011	0.714

* *p* < 0.05. ** *p* < 0.01. *** *p* < 0.001.

## Data Availability

Data may be obtained from Prof. Menachem Ben-Ezra via email.

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
