# Peer review of "Association between COVID-19 and Seasonal Influenza Vaccines to Vaccine Hesitancy, Intolerance of Uncertainty and Mental Health"

_vaccines, 2023, doi:10.3390/vaccines11020403_

Round 1

Reviewer 1 Report

This paper is well-written, well-organized, and very interesting. It also provides important information for associations between covid-19 and seasonal influenza vaccines to vaccine hesitancy, intolerance of uncertainty, and mental health.

The scientific background and the evidence available in the literature are completely reported.The description of the results of other studies cited in the paper is fully reported. The method is the main strength of the paper.

The statistical methodology is feasible and the conclusions assumed aren't excessive in comparison with the parameters included in the analysis.

The limits of the study couldn’t be further discussed.

Nevertheless, the authors should clarify some points, as suggested below

  • In Line 102 authors mention the STROBE guidelines for observational studies. But they don’t cite the report that this is written.
  • In line 103  authors mention  “  We used Qualtrics platform to deploy the survey…”      It would be more appropriate, for the authors, to insert some information about Qualtrics.

Well done!!!!!!!!!

Author Response

Response to Reviewer 1 Comments

This paper is well-written, well-organized, and very interesting. It also provides important information for associations between covid-19 and seasonal influenza vaccines to vaccine hesitancy, intolerance of uncertainty, and mental health.

The scientific background and the evidence available in the literature are completely reported. The description of the results of other studies cited in the paper is fully reported. The method is the main strength of the paper.

The statistical methodology is feasible and the conclusions assumed aren't excessive in comparison with the parameters included in the analysis.

The limits of the study couldn’t be further discussed.

Nevertheless, the authors should clarify some points, as suggested below

Answer: We thank Reviewer #1 for the positive feedback as well as for the helpful suggestions for revising and improving our manuscript. We have revised the paper in accordance.

Point 1: In Line 102 authors mention the STROBE guidelines for observational studies. But they don’t cite the report that this is written.

Response 1:  We thank the reviewer for this important comment. We have corrected this.

Lines 102-106: "The STOBE guidelines are STrengthening the Reporting of OBservational studies in Epidemiology. These guidelines are considered the common practice for good epidemiological studies. Cited from: https://www.strobe-statement.org/" 

Point 2: In line 103  authors mention  “  We used Qualtrics platform to deploy the survey…”      It would be more appropriate, for the authors, to insert some information about Qualtrics.

Response 2: The manuscript was amended accordingly:

Lines 106-109:

"The Qualtrics platform is a widely used survey platform to run online surveys. Its ease of usage is convenient to both research teams and sample populations. It is a widely used platform, utilized by numerous academic institutions worldwide."

We once again thank Reviewer #1 for the valued comments that helped improve our paper. 

Reviewer 2 Report

The paper is very interesting but I have some observations: 

At page 2, line 59-60, the authors should describe the characteristics of specific scales regarding vaccine hesitancy

At page 3,line 145,  the authors should give more informations about  the Bonferroni post-hoc test

At page 14, line 230, the authors should explain better the different methodologies used to consider intolerance of uncertainty

At page 15, line 256, the authors should give a mention about their personal solutions to increase potential mental health outcomes regarding the vaccinations 

Author Response

Response to Reviewer 2 Comments

The paper is very interesting but I have some observations: 

Answer: We thank Reviewer #2 for the positive feedback as well as for the helpful suggestions for revising and improving our manuscript. We have revised the paper in accordance.

Point 1: At page 2, line 59-60, the authors should describe the characteristics of specific scales regarding vaccine hesitancy

Response 1:  We thank the reviewer for this important comment. We have corrected this.

Lines 60-63: " Such scales address several domains with regard to vaccine hesitancy, such as trust or mistrust of vaccine benefit, worries over unforeseen future effects, concerns about commercial profiteering, and preference for natural immunity [16-18]."

Point 2: At page 3,line 145,  the authors should give more information about the Bonferroni post-hoc test.

Response 2: The Bonferroni Correction is used in multiple comparison test as a post hoc correction when many independent or dependent variables are being statistically tested at the same time. This correction is addressing the problem of running many simultaneous tests is that the probability of a significant result increases with each test run. This post hoc test sets the significant cut off at α/n. For example, if we are running 20 simultaneous tests at α = 0.05, the correction would be 0.0025.

Point 3: At page 14, line 230, the authors should explain better the different methodologies used to consider intolerance of uncertainty

Response 3: The manuscript was amended accordingly:

Lines 233-237:

"There are two approaches to interpret and analyze IUS-12 scale scores. The first is calculating each subscale (prospective and inhibitory) and use them separately; We did not deploy this method in the current paper. For the purpose of the current study, we utilized and scored the IUS-12 as a whole, meaning a global measure of intolerance to uncertainty".

Point 4: At page 15, line 256, the authors should give a mention about their personal solutions to increase potential mental health outcomes regarding the vaccinations 

Response 4: The manuscript was amended accordingly:

Lines 267-271:

"Therefore, the results of the current study may be utilized by health policymakers to address campaigns against vaccine hesitancy by focusing on its potential mental health outcomes, such as depression and anxiety, and suggest ways to improve intolerance of uncertainty by those individuals who possess anti-vaccine attitudes."

We once again thank Reviewer #2 for the valued comments that helped improve our paper.